# Effect of Rotary Speed on Soil and Straw Throwing Process by Stubble-Crushing Blade for Strip Tillage Using DEM-CFD

**Yiwen Yuan** [1], **Jiayi Wang** [1], **Xin Zhang** [2] and **Shuhong Zhao** [1,*]

1   College of Engineering, Northeast Agricultural University, Harbin 150030, China; yuanyw@neau.edu.cn (Y.Y.)
2   Heilongjiang Academy of Agricultural Machinery Engineering Sciences, Harbin 150081, China
*   Correspondence: shhzh@neau.edu.cn

**Abstract:** Strip tillage is a widely used land preparation approach for effective straw management in conservation agriculture. Understanding the dynamic throwing process during the stubble-crushing operation has important implications for seedbed preparation. However, the airflow generated by the high-speed rotation of stubble-crushing blades has yet to be considered. We established a coupled DEM-CFD simulation model and explored the dynamic motion of soil particles varied with their initial depth (at 0, 20, 40, 60, 80 mm depth) and surface straw under different blade rotary speeds (270, 540, 720, and 810 rpm) based on the model. The coupled model simulation results were proved to be well correlated with the field test results by the field high-speed camera test. The simulation results showed that airflow had a significant effect on the longitudinal displacement of straw ($p < 0.05$). An increase in rotary speed could increase the longitudinal and lateral throwing displacement of soil particles and straw and increase the blade–soil–straw interaction, while there was no directional effect on the vertical motion. The lateral movement of soil particles decreased with increasing soil particle depth. The stubble-crushing operation allowed the exchange of deep and shallow soil layers, as well as the burial of straw. Plain, straight stubble-crushing blades with a rotary speed of 540 rpm were able to form the optimal seeding row with a width of 80.86 mm. The simulation results were useful for assessing the design solutions of high-speed rotational tools and evaluating seedbed preparation practices.

**Keywords:** stubble-crushing; plain straight blade; DEM-CFD; dynamic motion; seedbed preparation; strip tillage

## 1. Introduction

Conservation agriculture is an idea with expanding global adoption due to its potential for soil conservation and increased crop productivity [1–3]. The core principles of conservation agriculture include adequate retention of straws, minimum soil disturbance, and diversified crop rotation [4], the first two leading to increased soil surface straw loads [5]. In conservative agricultural systems, excessive straw remaining in the fields has been a yield-limiting problem. Therefore, mechanized technology for managing straws is necessary to ensure the quality of no and minimum tillage and straw returning operations in a conservation agriculture system. Northeastern China is the main crop-producing area of the country [6], and maize is one of the most important grain crops, which is cultivated in areas of $12.06 \times 10^6$ ha, while generating an enormous amount of straw each year.

Under the straw returning and mulching management model, the terrain is complex, with a large amount of straw and stubble. Therefore, treating straw is crucial after fall harvest or before planting in the spring, or to plough the land as little as possible. Strip tillage, the procedure of placing seed into narrow furrows and limiting soil inversion, is relatively new, having been first evaluated in the early 1990s [7]. This tillage mode, shown in Figure 1, confines tillage to a narrow strip where the crop will be planted and divides the cropping system into two distinct adjacent rows: tilled and untilled [8]. Tillage within

the crop row can create a finer seedbed [9], and the untilled zone between rows maintains no-tilled states. Compared with traditional integrated tillage and rotational tillage, strip tillage has less soil disturbance, higher operational efficiency for soil water storage and moisture conservation, and improvement of the ground temperature of the seedbed, which could significantly improve the quality of subsequent seeding [10]. A high-quality strip crop row improves seed emergence rate, shortens emergence time, and then improves crop yield. Strip tillage, according to Al-Kaisi and Yin [11] and Gathala et al. [12], could help intensify dry land cropping by reducing land fallow and ensuring adequate yields, while also reducing greenhouse gas emissions and improving energy efficiency.

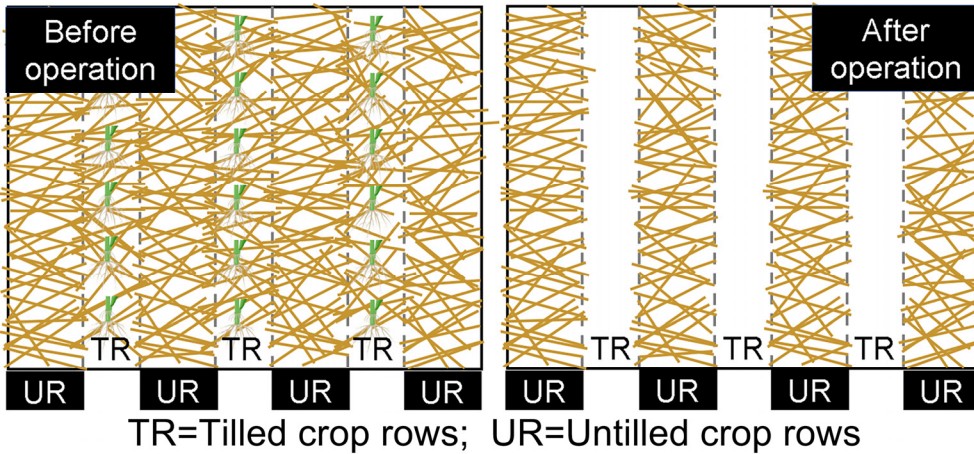

**Figure 1.** Effectiveness of maize straw mulch strip-till stubble-crushing operation under-conservation agriculture system.

Evidence indicates the critical soil engaging component is the blade [13]. In strip-till under-conservation agriculture system, stubble-crushing blades cut through crop straw and soil during the stubble-crushing operation. Soil and straw displacement are the highlights of the relevant literature. Fang et al. [14,15] analyzed the soil macroscopic and fine motion behavior and the displacement of rice and wheat straw in the rototilling process, which helped us to understand the interaction of the rotary blade, soil, and straw. To address the issue in the working process of strip tillage seeders, Zhao et al. [16] examined the effect of different edge–curve geometries on soil disturbance characteristics. Overthrowing, on the other hand, may degrade the quality of rotary strip tillage seedbed furrows. The effectiveness of seed germination was shown to be decreased when too much dirt was dumped out of the furrow, leading to a lack of soil cover over the seeds and a propensity to coat the furrow (resulting in a thin layer of compacted soil). In stubble-crushing operations, the high-speed rotation of the driven blades creates a non-negligible airflow around them. Airflow has a great influence on the motion of soil and straw, while airflow generated by high-speed rotational tools has not been considered. Under the condition of straw returning, the effect of soil and straw throwing separation affects the sowing zone environment and increases working resistance and power consumption. Hence, it is of great significance to explore the mechanism of throwing separation of soil and straw.

The numerical method has demonstrated considerable advantages over analytical and experimental methods in investigating soil–tool interaction problems, being less time-consuming and more capable of studying the complex geometry of soil-engaging tools [17]. It is becoming common practice to use the discrete element method (DEM) for numerical modeling in order to accurately predict soil dynamics [18–20]. A soil–tool–straw interaction DEM model was developed using the discrete element method to reproduce the soil bin test [21]. The DEM model was validated using experimental results by simulating the effect of narrow point opener geometry on soil disturbance and cutting [22]. Zhao et al. [16] used the discrete element method (DEM) to examine the effects of different edge–curve geometries on torque requirements and soil disturbance characteristics. Fang et al. [14]

studied the macro- and meso- movements of soil particles during rotary tillage with the help of DEM simulation. The flow field generated by the high-speed rotation of blades cannot be simulated by DEM softwares, despite the fact that it can be used to obtain some microscopic data that cannot be collected in a real test. Computational fluid dynamics (CFD) has also become a widely used and highly valued engineering tool among agricultural researchers for simulating fluid flow, the characterization of flow fields, and revealing fluid phase mechanisms for sustainable development [23]. To describe detailed dynamic information, including particle velocity, instantaneous forces acting on each particle by airflow, and interaction between gas–solid phases, discrete element method and computational fluid dynamics (DEM-CFD) techniques have been combined [24]. By the DEM-CFD coupling method, the purpose of forming a flow field by simulating the high-speed rotation of blades can be achieved.

Bent blade and plain straight blade are the two main types used for strip tillage. For a variety of reasons, the use of bent blades might result in an increased amount of soil being thrown out in untilled crop rows [13]. A plain straight blade is more compatible with the requirements of strip tillage to reduce soil disturbance. Therefore, the objective of this study was to investigate the effect of rotary speed and airflow on soil and straw throwing processes in plain straight blade operation based on the fluid–solid coupling simulation model of the strip-till under-conservation agriculture system.

## 2. Materials and Methods

### 2.1. Development of Coupling Contact Model

The discrete element method simulates the process of motion propagation in a collection of particles. The particle motion will inevitably cause mutual collisions between the particles, and the particles will inevitably generate force between them. A contact model describes how elements behave when they come into contact with each other. The discrete element method, which is based on contact mechanics and elastoplastic analysis of granular materials under quasi-static circumstances, relies heavily on the contact model. Particle trajectories are calculated analytically from forces and moments applied to them, which are then determined by the trajectory of those particles during testing. To increase the accuracy of the simulation findings, distinct contact types must be constructed for different simulation objects.

Hertz–Mindlin (no slip) is the default contact model added between soil and blade. In this model, both normal and tangential forces have damping components, where the damping coefficient is related to the recovery factor. The tangential friction follows Coulomb's law of friction, and the rolling friction is implemented as a directional constant torque model independent of the contact.

To describe the cohesion of agricultural soils, in addition to the Hertz–Mindlin (no slip) contact model between soil particles [25–27], considering the strong cohesion properties between clay particle, the Hertz–Mindlin with bonding (HMB) contact model was added [28–30]. This contact model is applied to bond the particles to simulate the surface straw as well as the root stubble model after the straw was crushed and returned to the field. In the HMB contact model, a "binder" is added between the particles to bond them together. This "binder" is capable of withstanding normal and shear displacements, and the bond is broken when normal and shear stresses exceed their limits. The moisture content is determined by the bonded radius and density under this model, shown in Figure 2, satisfying the following relationship:

$$\omega_s = \frac{\rho_W V_w}{\rho_s V_s} \tag{1}$$

$$V_s = \frac{4}{3}\pi R_P^3 \tag{2}$$

$$V_w = \frac{4}{3}\pi R_B^3 - \frac{4}{3}\pi R_p^3 \tag{3}$$

where $\omega_s$, $V_s$ and $\rho_s$ are, respectively, moisture content, volume (mm$^3$), and density (g·mm$^{-3}$) of soil particle. $V_w$ and $\rho_w$ represent the volume (mm$^3$) and density (g·mm$^{-3}$) of water. $R_B$ and $R_P$ are the bonded radius (mm) and soil particle radius (mm), respectively. Under the condition of known water density, the soil moisture content and soil density were measured, and the water volume was obtained by Equation (1). Furthermore, the soil particle radius was measured, and the bonded radius was calculated by Equation (3). Therefore, the definition of particle density and particle bonded radius in the HMB model indirectly reflected the particle moisture content.

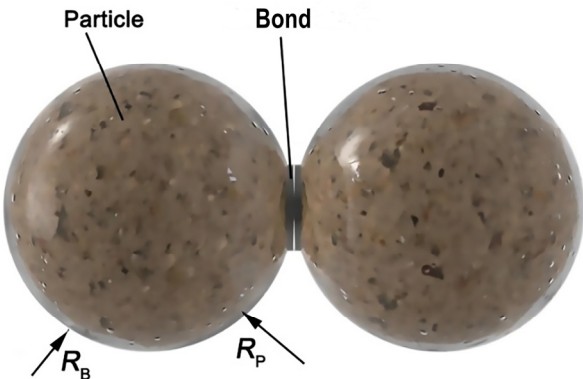

**Figure 2.** The bond relationship between the wet soil particles.

A Hertz–Mindlin with the JKR (Johnson–Kendall–Roberts) cohesion contact model, based on the Johnson–Kendall–Roberts theory [31], was added between the stubble and soil [32–36] to simulate crop growth.

The model was imported into the CFD software (Fluent 17.0, ANSYS, Canonsburg, PA, USA) module to establish the interaction between the stubble-crushing blade and airflow. A workflow of the CFD-DEM coupling process is presented in Figure 3. In the CFD-DEM coupling model, no complex energy transfer processes are involved, so only mass conservation and momentum conservation equations (Navier–Stokes equations) are used in this simulation to describe the continuous gas phase. The equations for the gas-phase mass and its momentum conservation, respectively, are given as follows:

$$\frac{\partial \varepsilon \rho}{\partial t} + \nabla \cdot \rho \varepsilon \vec{u} = 0 \tag{4}$$

$$\frac{\partial \varepsilon \rho \vec{u}}{\partial t} + \nabla \cdot (p\varepsilon \vec{u}\vec{u}) = -\varepsilon \nabla p + \nabla \cdot \left(\varepsilon \vec{\tau}\right) + \rho \varepsilon g - \vec{F}_{pf} \tag{5}$$

where $\rho$ is the fluid density, $\varepsilon$ is the fluid volume fraction without the CFD mesh, $\vec{u}$ is the fluid velocity, $p$ is viscous stress tensor, and $\vec{F}_{pf}$ is total force density due to the presence of particles.

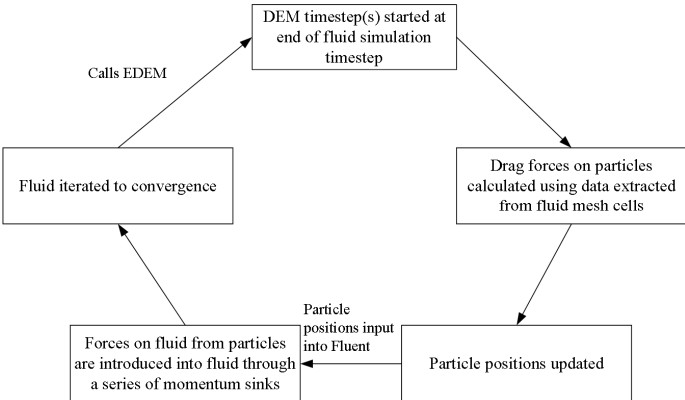

**Figure 3.** Flow chart of EDEM-FLUENT coupling processes and information exchange between two solvers.

The k-ε turbulence model was developed based on Boussinesq's assumption of isotropic eddy viscosity coefficient—Reynolds stress is proportional to the average velocity gradient. The working condition was turbulent, and the RNG (renormalization group) k-ε turbulent flow energy dissipation rate model was used [37–40]. The transport equation and turbulent viscosity equation were

$$\frac{\partial(\rho k)}{\partial t} + \frac{\partial(\rho k u_i)}{\partial x_i} = \frac{\partial}{\partial x_j}\left[\alpha_k \mu_{\text{eff}}\frac{\partial_k}{\partial x_j}\right] + G_k + \rho\varepsilon_t \tag{6}$$

$$\frac{\partial(\rho\varepsilon_t)}{\partial\varepsilon_t} + \frac{\partial(\rho\varepsilon_t)}{\partial x_i} = \frac{\partial}{\partial x_j}\left[\alpha_s \mu_{\text{eff}}\frac{\partial_c}{\partial x_j}\right] + \frac{C_{1c}\varepsilon_t}{k}G_k - C_{2c}\rho\frac{\varepsilon_t{}^2}{k} \tag{7}$$

$$\mu_{\text{eff}} = \mu + \mu_t \tag{8}$$

$$\mu_t = \rho C_\mu \frac{k^2}{\varepsilon_t} \tag{9}$$

$$C_{1\varepsilon}^* = C_{1\varepsilon} - \frac{\eta(1-\eta/\eta_0)}{1+\beta_{k-\varepsilon}\eta^3} \tag{10}$$

$$\eta = (2E_{ij}E_{ij})^{\frac{1}{2}}\frac{k}{\varepsilon_t} \tag{11}$$

$$E_{ij} = \frac{1}{2}\left(\frac{\partial u_i}{\partial x_j} + \frac{\partial u_j}{\partial x_i}\right) \tag{12}$$

where $x_i$, $x_j$ are components of displacement in $i$ and $j$ directions; $\alpha_k$ $\alpha_\varepsilon$ are turbulent kinetic energy $k$ and dissipation rate $\varepsilon$ corresponding Prandtl number, $\alpha_k = \alpha_\varepsilon = 1.39$; $u_{\text{eff}}$ represents correction parameters; $u_i$, $u_j$ are components of velocity in $i$ and $j$ directions; $\varepsilon_t$ represents turbulent dissipation rate; $\mu$, $\mu_t$ are turbulent viscosity coefficient and viscosity; $E_{ij}$ is mainstream time average strain rate, and $\eta$ is strain rate; $C_{1\varepsilon}$, $C_{2\varepsilon}$, $C_\mu$ are empirical constant; $C_{1\varepsilon} = 1.42$, $C_{2\varepsilon} = 1.68$ and $C_\mu = 0.0845$; $\eta_0$, $\beta_{k-\varepsilon}$ represent constant.

### 2.2. Simulation Model Development and Parameters

To simulate the interaction between soil particles, maize straw, maize stubble, and plain straight blades, a simulation model was developed in EDEM 2018$^{\text{TM}}$ software (DEM Solutions, Edinburgh, UK), which consisted of blades and a soil bin containing maize straw and stubble model composition.

In order to ensure the spatial adequacy, a virtual soil bin model of 1500 mm long × 400 mm wide × 150 mm deep was created using default circular particles, which were made up of 4 mm radius (selected based on available computation time), and soil particles were randomly generated and settled to reach an equilibrium state in soil bin. Black soil (clay loam) in Northeast China was used for this modeling. The maize straw piece hollow model was created using the 3D modeling software SolidWorks 2016 (Dassault Systemes, Suresnes, France), imported into EDEM 2018[TM], and the position of the straw model was adjusted by Reposition under Geometry. Straw models were filled by default circular particles of 1.5 mm in radius. Six straw pieces were placed on the soil surface as straw tracers. An additional 110 straw pieces, which were straws with lengths of 50, 75, and 100 mm and radii of 10, 12.5, and 15 mm, respectively (suggested by [28,34,41]), were randomly spread on the surface to achieve the same straw cover in mass as in the validation tests. The geometric model surface of stubble was designed as suggested by Wang et al. [42]. Four maize root stubbles, each with a spacing of 30 mm, were generated by combining the solid and virtual stubble geometry model and placed uniformly in the soil bin. The geometric models of the straw and stubble overlapped with the virtual surface, and the geometric model surface was deleted when the particles filled the geometric model (the soil bin, straw, and stubble models in Figure 4a). The plain straight rotary blade used for strip tillage was adopted in this study, as shown in Figure 5. A three-dimensional model of the stubble-crushing blade (density of 7865 kg·m$^{-3}$) was established in the Catia V5 software (Dassault Systemes, Suresnes, France), and then imported into the ICEM software for dynamic meshing, and the dynamic meshed stubble-crushing blade was imported into the EDEM 2018[TM] to establish the interaction model of the coupling field. The EDEM 2018[TM] was coupled with Fluent 17.0 using the udf coupling interface, and a flow field area of 1500 × 400 × 1000 mm in length × width × height was created in Fluent 17.0 by adding a natural wind of 3 m·s$^{-1}$ in the direction opposite to that of the stubble-crushing blade movement (Figure 4b). Three cylindrical air layers of 230 mm radius and 6 mm thickness were added around the stubble-crushing blade so that the blades were placed in them. The parameters to be determined for the soil–straw–stubble–blade interaction model are shown in Table 1.

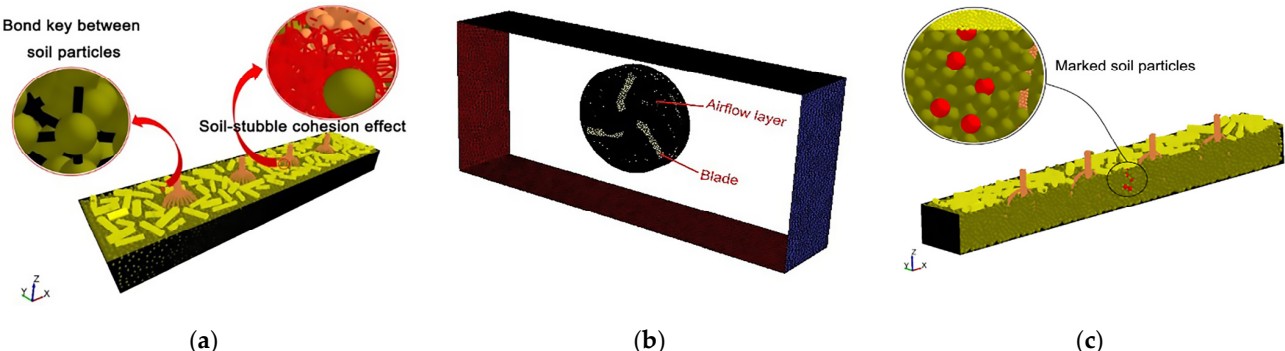

(a) (b) (c)

**Figure 4.** DEM-CFD coupling model: (**a**) soil, straw, and stubble model; (**b**) flow field model; (**c**) layout of marked soil particles single rental mark.

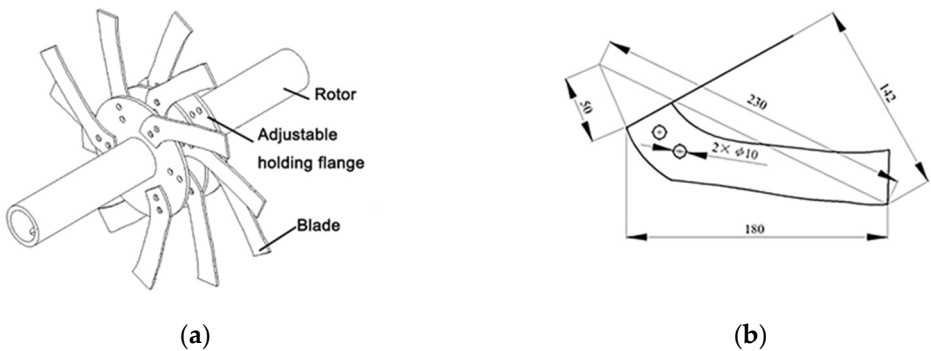

(**a**)                                    (**b**)

**Figure 5.** Soil-engaging tool of stubble-crushing tested: (**a**) model of the field validation experimental rotor with twelve blades fitted onto adjustable holding flanges; (**b**) geometric parameters of the plain straight stubble-crushing blade.

**Table 1.** Simulation model key parameters.

| Objects | Parameters and Units | Values |
|---|---|---|
| Soil | Poisson's ratio | 0.41 |
| | Shear modulus (MPa) | $1.24 \times 10^6$ |
| | Density (kg·m$^{-3}$) | 2150 |
| | Coefficient of rolling friction | 0.2 |
| | Coefficient of static friction | 0.3 |
| | Coefficient of restitution | 0.6 |
| | Critical normal stress (kPa) | 200 |
| | Critical shear stress (kPa) | 68 |
| | Normal contact bond stiffness (N·m$^{-1}$) | $3.4 \times 10^8$ |
| | Shear contact bond stiffness (N·m$^{-1}$) | $1.5 \times 10^8$ |
| | Moisture | $20 \pm 1\%$ |
| Straw | Poisson's ratio | 0.4 |
| | Density (kg·m$^{-3}$) | 241 |
| | Shear modulus (MPa) | 1 |
| | Shear modulus (MPa) | 0.6 |
| | Coefficient of static friction | 0.01 |
| | Coefficient of static friction | 0.3 |
| | Critical normal stress (kPa) | $8.72 \times 10^3$ |
| | Critical shear stress (kPa) | $7.5 \times 10^3$ |
| | Normal contact bond stiffness (N·m$^{-1}$) | $9.6 \times 10^6$ |
| | Shear contact bond stiffness (N·m$^{-1}$) | $6.8 \times 10^6$ |
| Stubble | Bonded radius (mm) | 1.79 |
| | Poisson's ratio | 0.33 |
| | Density (kg·m$^{-3}$) | 107.64 |
| | Shear modulus (MPa) | 6.293 |
| | Coefficient of restitution | 0.6 |
| | Coefficient of rolling friction | 0.21 |
| | Coefficient of static friction | 0.573 |
| | Bond radius (mm) | 1.7 |
| | Critical normal stress (kPa) | 500 |
| | Critical shear stress (kPa) | 500 |
| | Normal contact bond stiffness (N·m$^{-1}$) | $1.034 \times 10^6$ |
| | Shear contact bond stiffness (N·m$^{-1}$) | $1.034 \times 10^6$ |
| Blade | Shear modulus (MPa) | $7.9 \times 10^4$ |
| | Poisson's ratio | 0.3 |
| Soil-stubble | Coefficient of restitution | 0.60 |
| | Coefficient of static friction | 0.60 |
| | Coefficient of rolling friction | 0.02 |
| | Surface energy (J·m$^{-2}$) | 10 |
| Soil-blade | Coefficient of restitution | 0.6 |
| | Coefficient of static friction | 0.313 |
| | Coefficient of rolling friction | 0.107 |

**Table 1.** *Cont.*

| Objects | Parameters and Units | Values |
|---|---|---|
| Straw-blade | Coefficient of static friction | 0.3 |
| | Coefficient of rolling friction | 0.01 |
| Stubble-blade | Coefficient of static friction | 0.6 |
| | Coefficient of rolling friction | 0.02 |

The time step is the amount of time between iterations in the EDEM 2018$^{\text{TM}}$ (computational particle and contact between particles and geometry). In the process of soil particle generation, the fixed time step was $8.2 \times 10^{-5}$ s (less than 20% of the Raileigh time step [28]). The Rayleigh time step is the time taken for a shear wave to propagate through a solid particle [43], which was set to $5.47 \times 10^{-4}$ s. The soil particles generation time was set from 0 to 6 s, and the settling time was 1 s. The generation times of the maize straw pieces and root stubbles were 1 s, respectively. Based on this, the airflow–soil solid coupling field of the stubble-crushing blade operation was established.

### 2.3. Simulation Experiment

The two-factor experiment was designed as a randomized complete block with four replications [44,45]. The experimental factors were the rotary speed of the blade and the presence or absence of airflow. In addition, the blade rotary speeds were set to 270, 540, 720, and 810 rpm (including the high and low output shaft cutter roll speeds of the tractor). The airflow layer interacting with the blade (the airflow rate was directly output in the Fluent17.0 software) was set in the Fluent17.0 software, and the Coupling Server module in the EDEM 2018$^{\text{TM}}$ was started for coupling simulation. At the same time, a separate simulation of the operation of the blade was performed in the EDEM 2018$^{\text{TM}}$. The experimental indicators were the lateral displacement of soil particles and straw particles (perpendicular to the driving direction of the tractor), vertical displacement, longitudinal displacement (opposite to the driving direction of the tractor), lateral velocity, longitudinal velocity, and vertical velocity. The average value of the indicators was used as the final experimental result of each group of experiments.

### 2.4. Model Monitoring

Throughout the simulation, soil particles were monitored by coloring them red. In the study of soil particles, since the operating depth of the stubble-crushing blade was 80 mm, the soil particles at each depth were marked in layers, which were 0, 20, 40, 60, and 80 mm, and four locations were randomly marked at each depth, and the straw model was randomly marked in four places on the surface, as shown in Figure 4c. The X, Y, and Z directions were defined as longitudinal (opposite to the working direction), lateral (perpendicular to the working direction), and vertical directions, respectively. The dynamic motion (displacement and velocity) of the marker in each of the three directions during the blade stabilization operation was output using the built-in post-processing module in the EDEM 2018$^{\text{TM}}$. The simulation process is shown in Figure 6.

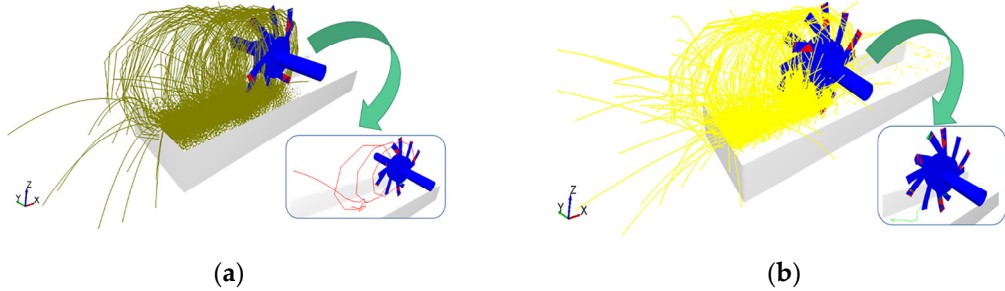

(**a**)          (**b**)

**Figure 6.** Simulation motion trajectory: (**a**) soil particles; (**b**) straw.

*2.5. Field Experiment*

In order to obtain the airflow velocity around working-blade and validate DEM-CFD model, the field test was conducted at the research farm (44°04′~46°40′ N and 125°42′~130°10′ E) of the Northeast Agricultural University located in the Heilongjiang Province of northeastern China in October 2020. This region has a typical temperate continental monsoon climate and shows a continental climate, with a short hot period during summer and a long cold period during winter. The mean annual precipitation is 569.1 mm and more than 60% of precipitation occurs from June to September. The average annual potential evapotranspiration is 1009 mm (1971–2000) and markedly exceeds annual precipitation. Soil in the northeast is typically black soil with a clay loam texture containing 36.0% sand, 24.5% silt, and 39.5% clay [46]. The average soil moisture content of the cultivation layer within 80 mm of the experimental site was 20 ± 1%, and the average moisture content of maize root stubble is 184.3% (d.b). Prior to the tillage tests, the field had been used for maize with a ridged culture under a conservation tillage system. The distance between two adjacent ridges was 650 mm. Before the test, straws were returned to the field. The average stubble height on the ground was 110 mm, and the straw coverage was 1.031 kg·m$^{-2}$. The average moisture content of soil, straw, and stubble was 20.5%, 19.4%, and 43.9%, respectively.

The field test is shown in Figure 7. A single test rotor was fitted with twelve blades and set to cut maize straws and stubbles on the ground. A double-row stubble-crushing implement fitted with two rotors assembly was pulled by a Benye 454 Tractor (45 hp, Ningbo Benye Tractor & Automobile Manufacturing Co., Ningbo, China). The forward travel speed was kept constant at 0.56 m·s$^{-1}$, as was the rotor rotational speed at 810 rpm (equivalent to the rotational speed of the high-grade output shaft of the tractor). The test was repeated four times. Each test area was divided into an experimental area of 10 m and a preparation area of 5 m. The equipment was driven into the experimental area when the tractor traveling speed was kept constant at 0.56 m·s$^{-1}$ in the preparation area.

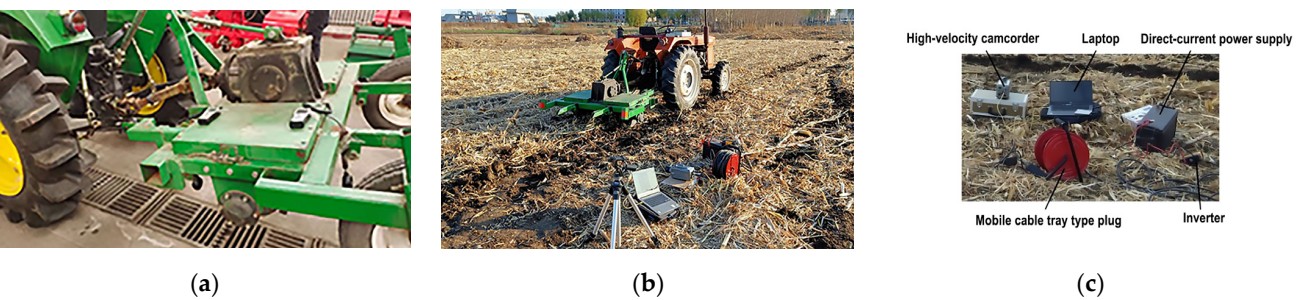

(**a**) (**b**) (**c**)

**Figure 7.** Apparatus of field experiment: (**a**) actual installation of air volume meter; (**b**) experimental devices; (**c**) electrical device connection.

Since the soil and straw thrown during the operation would hit the wind meter, we measured the wind speed data before the official test. A digital fractional wind meter (manufactured by Sigma Instruments Group Ltd., Hong Kong, with an accuracy of 0.001 m·s$^{-1}$) was installed on the stubble-crushing implement frame prior to the field experiment to measure and record the wind speed around the blades during the idling operation. During the operation of stubble-crushing blades at high-speed, blades can cut maize stubble, straws, and soil. The patterns of cutting and throwing are not possible to observe with the naked eye. Consequently, following Lee et al. [47] and Matin et al. [48], a high-velocity camcorder was used to catch symbolism-permitting representation of these examples. The video camera (Phantom V5.1, Vision Research Inc, Wayne, NJ, USA) was fitted to capture the images (1200 frames·s$^{-1}$, 1024 × 1024 dpi) of the process of throwing actions that help validate the simulation test results. The images were acquired on a laptop as RAW3 files using Phantom® Camera Control Application software (2.8, Vision Research Inc, Wayne,

NJ, USA) during the test runs. The images were analyzed with Promon Studio software (Vision Research Inc, Wayne, NJ, USA).

## 3. Results and Discussion

The dynamic displacement and velocity of soil particles with depths of 0, 20, 40, 60, and 80 mm and straw were extracted in EDEM 2018$^{TM}$, and the average velocity and displacement of four random soil particles at each depth and four random straws at the surface in different directions were taken for treatment. The processed experimental data were subjected to ANOVA, as shown in Table 2.

**Table 2.** Analysis of variance (F-test) of measured data.

| Source of Variation | Degree of Freedom | Soil *F*-Value | | | | | |
|---|---|---|---|---|---|---|---|
| | | X-Displacement | Y-Displacement | Z-Displacement | X-Velocity | Y-Velocity | Z-Velocity |
| Rotary speed, A | 3 | 1.434 ns | 0.153 ns | 5.449 ** | 3.696 ** | 0.041 ns | 0.126 ns |
| Airflow, B | 1 | 1.818 ns | 1.112 ns | 0.337 ns | 2.496 ns | 0.177 ns | 0.074 ns |
| AB | 3 | 0.823 ns | 0.096 ns | 0.322 ns | 0.078 ns | 0.027 ns | 0.101 ns |
| Error | 32 | | | | | | |
| **Source of Variation** | **Degree of Freedom** | **Straw *F*-Value** | | | | | |
| | | X-Displacement | Y-Displacement | Z-Displacement | X-Velocity | Y-Velocity | Z-Velocity |
| Rotary speed, A | 3 | 1.811 ns | 1.102 ns | 0.028 ns | 0.232 ns | 0.042 ns | 0.011 ns |
| Airflow, B | 2 | 3.830 * | 1.379 ns | 0.113 ns | 0.653 ns | 0.228 ns | 0.001 ns |
| AB | 2 | 0.247 ns | 0.003 ns | 0.074 ns | 0.088 ns | 0.009 ns | 0.027 ns |
| Error | 152 | | | | | | |

'*' means significantly different at $p < 0.05$; '**' means significantly different at $p < 0.01$; 'ns' means not significantly different at $p > 0.05$.

### 3.1. Testing Results and Model Validation

The field test airflow velocity was compared with the simulated airflow velocity. From the results of the field test, the airflow velocity around stubble-crushing blades was in the range of 0.602~4.988 m·s$^{-1}$ during the operation, and the airflow velocity in the simulation test was in the range of 0.613~4.85 m·s$^{-1}$. The airflow velocity set in the simulation test was basically the same as the surrounding airflow velocity during the operation of the plain straight stubble-crushing blade in the field test.

As obtained from the shooting results taken from the field high-speed camera test, the soil particles and straw on the ground surface were mainly thrown in the longitudinal direction. The longitudinal displacement and velocity of soil and straw were obtained in the DEM and the DEM-CFD coupled field, respectively. The correlation between simulated and measured values is given in Figure 8. The results obtained from the coupled models all lie above y = x. This indicates that the airflow action can increase the longitudinal displacements and velocities of the soil and straw, especially straw, which will accelerate the throwing motion and lead to an increase in the throwing range. The results in Figure 8a,c,e,g show that a better correlation was obtained between the DEM-CFD simulated and measured longitudinal motions, while a fair correlation was obtained between the measured and DEM simulated longitudinal motions (shown in Figure 8b,d,f,h). The correlation between the DEM simulated and measured straw longitudinal velocity was extremely poor. This meant that airflow did affect the accuracy of simulation results. Overall, the simulation results of the DEM-CFD coupled model showed better correlation with the results of the field tests compared with those DEM simulation model. Thus, the coupling simulation test can simulate the soil and straw thrown trend during operation.

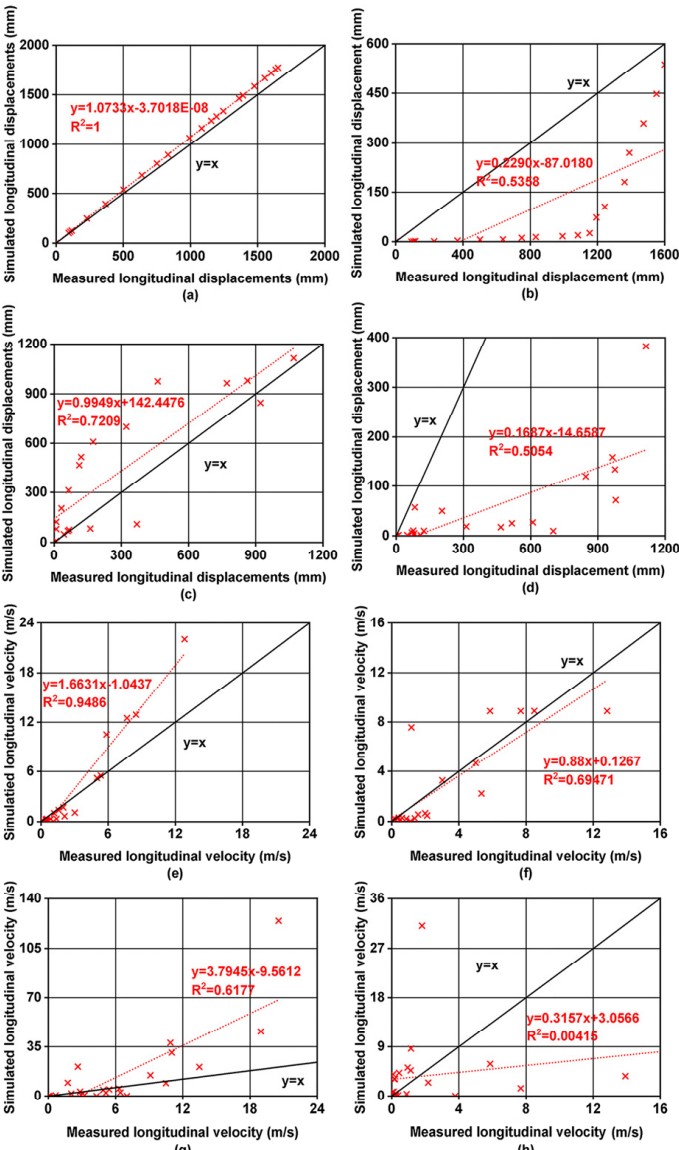

**Figure 8.** Correlation of longitudinal motion between the measured and simulated: (**a**,**b**) soil displacement in DEM-CFD and DEM model; (**c**,**d**) straw displacement in DEM-CFD and DEM model; (**e**,**f**) soil velocity in DEM-CFD and DEM model; (**g**,**h**) straw velocity in DEM-CFD and DEM model.

The overall determination coefficient ($R^2$) value of the longitudinal motion of straw is lower than that of soil due to the instability of straw throwing during stubble-crushing blade operation. The accuracy of the longitudinal motion results in the DEM-CFD simulation of the soil was higher than that of the straw, with an average relative error of 7.5% for the former and 17.8% for the latter. The results of the longitudinal displacement of soil and straw were more accurate, which was attributed to the fact that the extraction of velocity during the software processing was based on the change of displacement, which was the average velocity at the stage time instead of the instantaneous velocity. This had some deviation from the actual velocity variation, but it could reflect the velocity variation tendency of straw and soil and provide data support for the test results. On the whole, the model had an average relative error of 7.3% as compared to the testing results, which was lower than the 24.9% relative error of the analytical models in predicting soil forward displacement, as proposed by Fang et al. [14]. Therefore, the model was considered to be reasonably accurate in the simulation of soil–tool–straw interaction with airflow conditions.

### 3.2. Simulation of Soil and Straw Dynamics Displacements

### 3.2.1. Longitudinal Displacement

From the simulation test results (Figure 9), as the rotary speed of stubble-crushing blades increased from 270 to 540 rpm, the average longitudinal displacement of soil particles increased by 16.6% and that of straw by 89.9%. As the stubble-crushing blade speed increased from 540 to 720 rpm, the average longitudinal displacement of soil particles increased by 19.9%, while the average longitudinal displacement of straw increased by 106.8%. From 720 to 810 rpm, the average longitudinal displacement of soil particles increased by 17.3% and that of straw by 98.2%. Fang et al. [49] also observed that straw forward displacement was significantly larger than that of soil. Accordingly, increasing stubble-crushing blade rotary speed would increase the longitudinal throwing of soil and straw, which was the main throwing direction of soil and straw in stubble-crushing operations. According to Hendrick and Gill [50], the rotary speed of the blade influences soil throwing by affecting the soil acceleration. An increase in rotary speed results in a greater acceleration of soil particles, leading to more throwing.

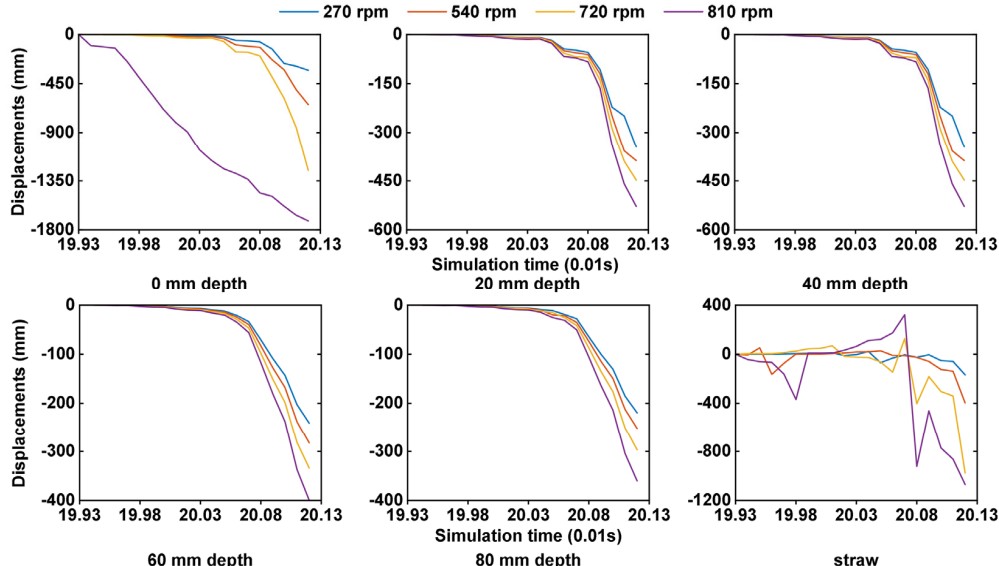

**Figure 9.** Longitudinal displacements of soil particles at different depths and surface straws at different rotary speeds.

The average longitudinal displacement of the soil particles decreased by 13% when the soil depth increased from 0 to 20 mm, by 13.7% when the soil depth increased from 20 mm to 40 mm, by 15.1% when the soil depth increased from 40 mm to 60 mm, and by 8.8% when the soil depth increased from 60 mm to 80 mm. Therefore, with the increase in soil depth, the longitudinal displacement of soil particles gradually decreased, and the longitudinal displacement of shallow soil was the largest, while the displacement of deep soil was the smallest. Therefore, most of the longitudinal soil thrown farther during the stubble-crushing operation was shallow soil.

As shown in Table 2, the presence or absence of airflow has a significant effect on straw displacement in X direction. Compared with no airflow, the longitudinal displacement of soil particles increased by 19.1% and the longitudinal displacement of straw increased by 335.9% under all of the working conditions with airflow. The effect of airflow on soil and straw throwing in stubble-crushing operation cannot be ignored. In summary, the lower the rotary speed of stubble-crushing blades, the smaller the loss of soil moisture, but this is not conducive to the separation of soil and straw throwing, so the maximum and minimum rotary speed were not the optimal choices.

### 3.2.2. Lateral Displacement

According to the simulation results (Figure 10), the average lateral displacement of soil particles increased by 31.5%, and the average lateral displacement of straw increased by 217.8% when the rotary speed of the stubble-crushing blades increased from 270 to 540 rpm. The average lateral displacement of soil particles increased by 14%, while the average lateral displacement of straw increased by 83.5%, when the rotary speed was raised from 540 to 720 rpm. The average lateral displacement of the soil particles increased by 20.3%, while the average lateral displacement of the straw increased by 61.3% when the rotary speed was raised from 720 to 810 rpm. Matin et al. [48] also reported an increasingly high amount of soil being thrown out of the furrow with greater rotary speed. Therefore, when the rotary speed rose, the lateral displacement of soil and straw also increased noticeably. This result showed that increasing rotary speed may greatly enhance the interaction between the blades, soil, and straw, aggravating the disturbance of soil and straw.

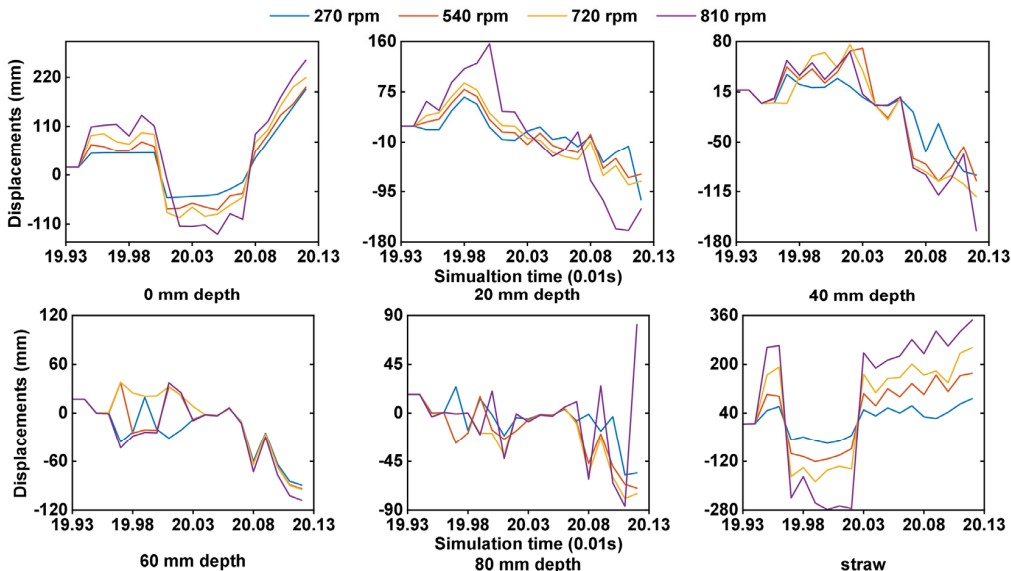

**Figure 10.** Lateral displacements of soil particles at different depths and surface straws at different rotary speeds.

The average lateral displacement of soil was reduced by 23.4% as the soil depth increased from 0 to 20 mm, and by 6.3% as the soil depth increased from 20 to 40 mm. The average lateral displacement of the soil particles also fell by 6.3% as the soil depth climbed from 40 to 60 mm. The average lateral displacement of the soil particles decreased by 32.9% when the soil depth was increased to 60 mm, and by 43.8% when it was increased from 60 mm to 80 mm. This is consistent with the findings of Fang et al. [14] that the lateral displacement of soil would be significantly reduced with an increase in soil depth. The lateral displacement of deep soil during the operation was minimally likely because the interaction between surface soil and lateral undisturbed soil was relatively weak, and as the soil depth increased, the force of lateral undisturbed soil on thrown soil increased. Therefore, the rotary speed should be suitably decreased to produce less lateral soil disturbance and lower power consumption.

### 3.2.3. Vertical Displacements

According to the test results (Figure 11), when the rotary speed of stubble-crushing blades was raised from 270 to 540 rpm, the average vertical displacement of the soil particles increased by 17.9% and that of straw decreased by 1.2%. However, when the rotary speed was raised from 540 to 720 rpm, the average vertical displacement of the soil particles decreased by 26.1% and that of the straw decreased by 1%. The average vertical displacement of the soil rose by 3.2% and that of the straw dropped by 2.1% when the rotary

speed was raised from 720 to 810 rpm. The vertical displacement of soil particles was not significantly affected by the rotary speed of the stubble-crushing blades. This is different from the higher rotary speed soil being thrown higher, as reported by Matin et al. [51], and may be related to the different soil types and working conditions. The findings suggest that increasing the rotary speed of the stubble-crushing blades will result in the disorderly vertical distribution of soil particles, while the final position of straw will definitely be lower than in the initial position. Thus, stubble-crushing operation may accomplish a specified soil covering of straw on the ground surface.

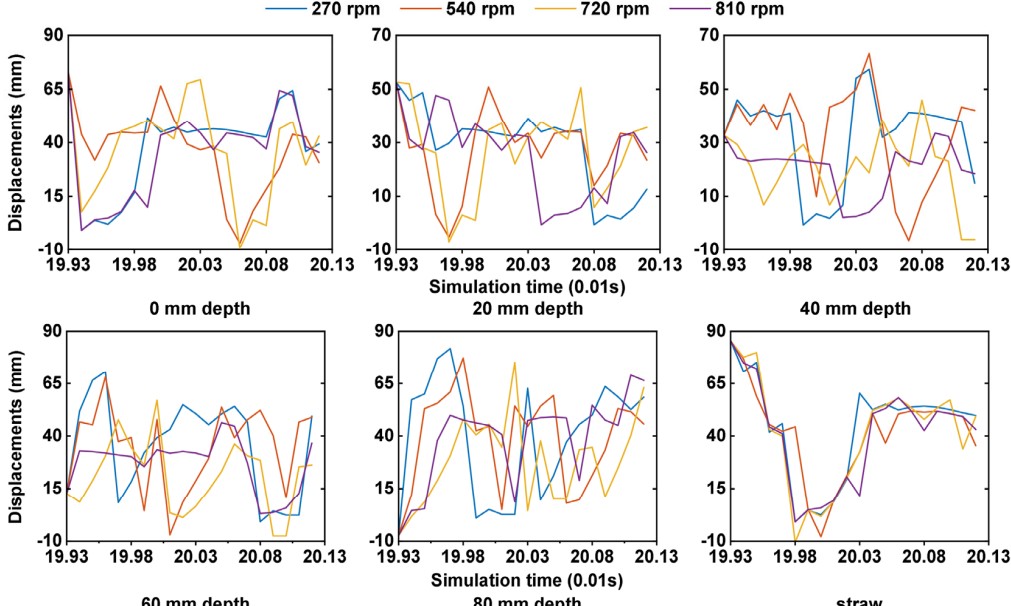

**Figure 11.** Vertical displacements of soil particles at different depths and surface straws at different rotary speeds.

When the soil depth increased from 0 to 20 mm, the average vertical displacement of the soil particles decreased by 24.2%. When the soil depth increased from 20 to 40 mm, the average vertical displacement of the soil particles decreased by 3.4%. When the soil depth increased from 40 to 60 mm, the average vertical displacement of the soil particles increased by 12.6%. Therefore, after the operation of stubble-crushing blades, the displacement of shallow soil will decrease and the displacement of deep soil will increase, resulting in the exchange of the deep and shallow soil layers and the change of soil layers. In conclusion, when the rotary speed of the stubble-crushing blades was 540 rpm with airflow action, the lateral displacement of the straw was analyzed. The statistics of the covered straw particles indicated that 12.5% of the total number of straws in the area were covered. Therefore, the majority of straws were disrupted in the lateral direction, and this operating condition aided in the establishment of a more effective seeding row.

To ensure a smaller lateral displacement of soil, the straight stubble blade angular speed should be reduced, but reducing blade rotary speed reduces the lateral displacement of straw. Considering the operation effect of the plain straight stubble-crushing blades, the better rotary speed is determined as 540 rpm, and the average unilateral lateral displacement of the soil is 40.43 mm, and the average unilateral lateral displacement of the straw is 102.46 mm. Therefore, the optimal sowing width of 80.86 mm can be formed, in theory.

### 3.3. Simulations of Soil and Straw Dynamics Velocities
#### 3.3.1. Longitudinal Velocities

According to Table 2, the rotary speed of blades has a substantial ($p < 0.05$) influence on the X-direction velocity of soil particles. When the stubble-crushing blades speed increased from 270 to 540 rpm, the average longitudinal velocity of soil particles increased

by 20% and the average longitudinal velocity of straw increased by 118.8%; when the rotary speed increased from 540 to 720 rpm, the average longitudinal velocity of the soil particles increased by 14.7% and the average longitudinal velocity of the straw increased by 108.9% (Figure 12); when the stubble-crushing blades rotary speed increased from 720 to 810 rpm, the average longitudinal velocity of soil particles increased by 16.8% and the average longitudinal velocity of straw increased by 38.8%, indicating that the rotary speed had an effect on the longitudinal velocities of the soil and the straw, with the effect on the longitudinal velocity of straw being more pronounced than that of the longitudinal velocity of soil. Increasing the rotary speed will boost the longitudinal throwing motions of the soil and the straw, which is also the primary throwing direction during stubble-crushing operations.

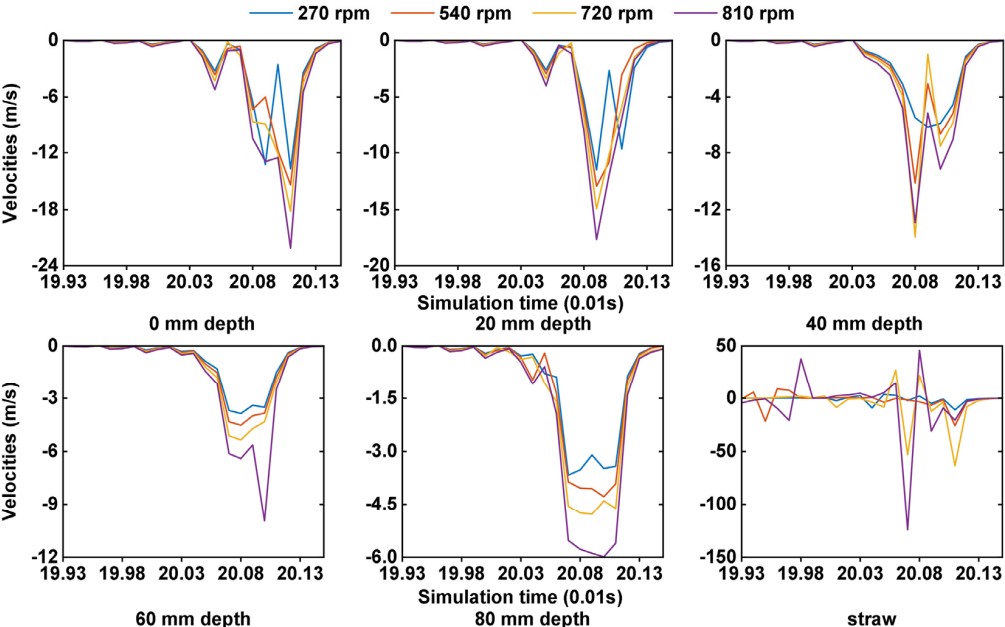

**Figure 12.** Longitudinal velocities of soil particles at different depths and surface straws at different rotary speeds.

The average longitudinal velocity of the soil particles fell by 19.7% as the soil depth increased from 0 to 20 mm, by 17% as the soil depth increased from 20 to 40 mm, by 23.7% as the soil depth increased from 40 to 60 mm, and by 8.3% as the soil depth increased from 60 to 80 mm. Therefore, the longitudinal velocity of the soil had a clear tendency to decrease with increasing soil depth, most likely because the interaction force between soil particles increased with the increasing soil depth and shallow soil impeded the vertical motion of deep soil, resulting in the weakening of the longitudinal motion of deep soil. Consequently, during stubble-crushing operations, shallow soil particles were mostly thrown at a greater distance, while deep soil particles were primarily thrown closer. To accomplish the separation motion of soil and straw, it is necessary to raise the blade rotary speed. This would, however, enhance the impact of blades on soil and straw, and the increased rotary speed would reflect the increased impact.

### 3.3.2. Lateral Velocities

The average lateral velocity of the soil particles increased by 60% and the average lateral velocity of the straw increased by 85.5% when the stubble-crushing blade rotary speed increased from 270 to 540 rpm (Figure 13). The average lateral velocity of the soil particles increased by 47.5% and the average lateral velocity of the straw increased by 79.2% when the rotary speed increased from 540 to 720 rpm. When the rotary speed was increased from 720 to 810 rpm, the average lateral velocity of the soil particles increased by 28.9% and the average lateral velocity of the straw increased by 74.7%. This shows that increasing

the rotary speed of the stubble-crushing blades will enhance the lateral throwing motions of soil and straw.

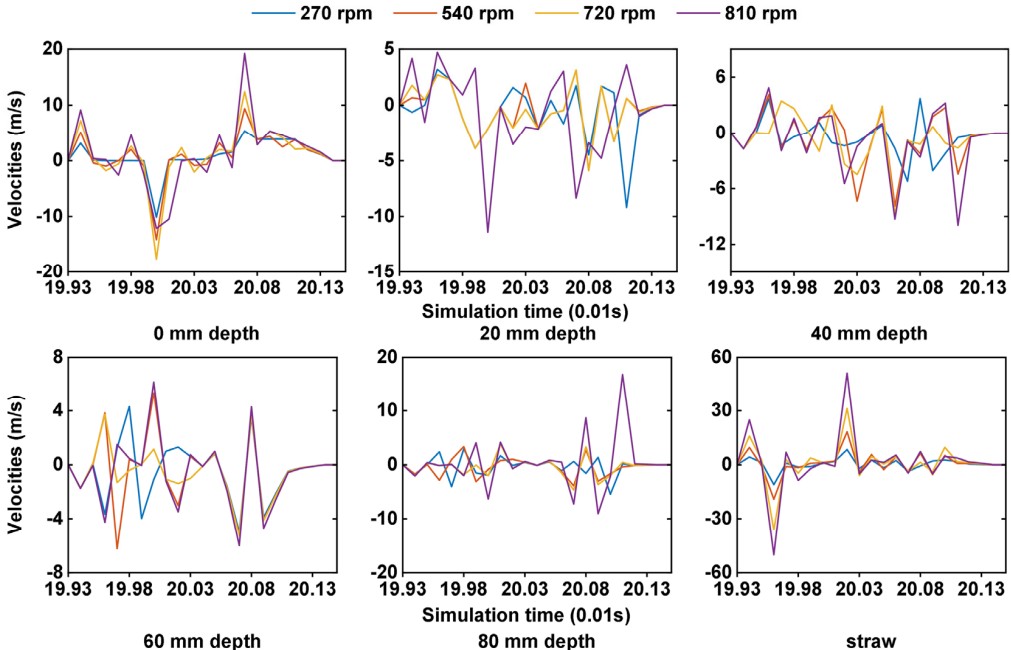

**Figure 13.** Lateral velocities particles at different depths and surface straws at different rotary speeds with or without airflow.

The average lateral velocity of the soil particles dropped by 29.3% as the soil depth increased from 0 to 20 mm, by 15.5% as the soil depth increased from 20 to 40 mm, by 43.23% as the soil depth increased from 40 to 60 mm, and by 67.8% as the soil depth climbed from 60 to 80 mm. Therefore, as the soil depth increased, the lateral velocity of the soil tended to decrease. This is likely because, as soil depth increases, the interaction force between the soil increases, which influences the trend of mutual motion between the soil particles and weakens the lateral motion of the soil. Therefore, the soil disturbed to the far sides during the stubble-crushing operation is mostly shallow soil, whereas the soil affecting the stubble-crushing operation region is often deep soil. In conclusion, if the lateral velocity of the soil is assured to be low (as a result of the weak contact between the stubble-crushing blades and the soil), the stubble-crushing blade rotary speed should be decreased, but this decreases the lateral velocity of straw. When the interaction between the stubble-crushing blades and the soil is poor, it is possible to create greater straw movement.

### 3.3.3. Vertical Velocities

According to the test results (Figure 14), the average vertical velocity of the soil particles increased by 15.5% and the average vertical velocity of the straw increased by 46.3% when the stubble-crushing blade speed was increased from 270 to 540 rpm. However, the average vertical velocity of the soil particles decreased by 75.5% and the average vertical velocity of the straw decreased by 9.5% when the rotary speed was increased from 540 to 720 rpm. The effect of the rotary speed on the vertical velocities of the soil and the straw was, thus, non-directional, and increasing stubble-crushing blades speed did not necessarily increase the vertical velocities of soil and straw.

The average vertical velocity of the soil particles decreased by 5.3% as the soil depth increased from 0 to 20 mm, by 79% as the soil depth increased from 20 to 40 mm, by 65.7% as the soil depth increased from 40 to 60 mm, and by 109.1% as the soil depth increased from 60 to 80 mm. Therefore, as the soil depth increased, the vertical velocity of the soil particles tended to decrease. This is likely because, as the soil depth increases, the interaction force between the soil increases, and the shallow soil will impede the vertical motion of

the deep soil, resulting in the weakening of the deep soil vertical motion. Consequently, during stubble-crushing operations, the shallow soil is primarily thrown to a higher height, whereas the deep soil typically returns to the surface after being thrown to a lower height.

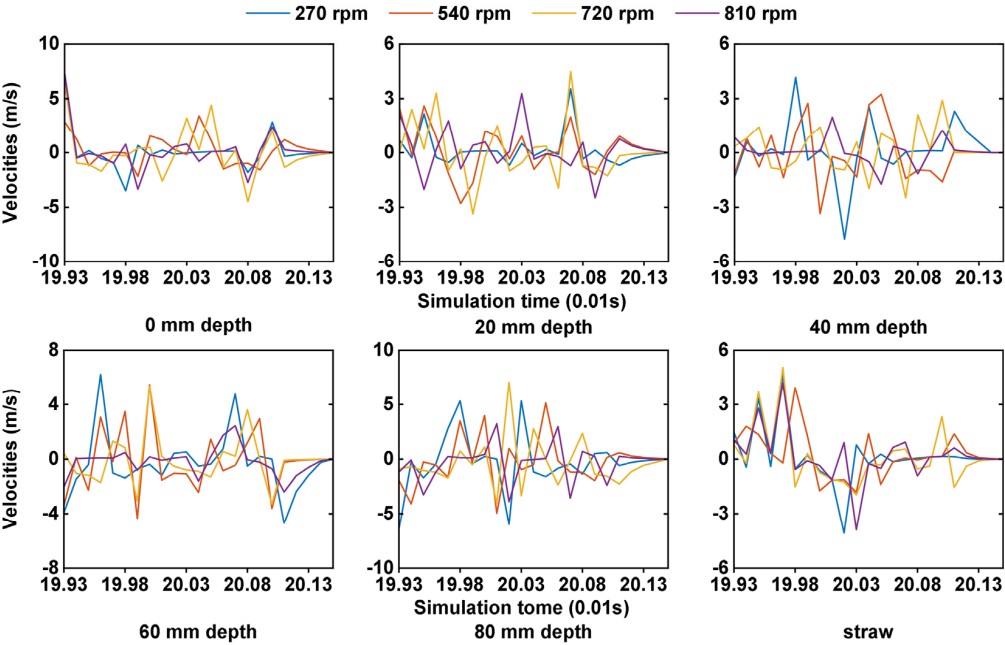

**Figure 14.** Vertical velocities particles at different depths and surface straws at different rotary speeds with or without air-flow.

## 4. Conclusions

In this study, a DEM-CFD model was developed to simulate the dynamic motion of soil and straw at different rotary speeds. The results of the field high-speed camera tests showed that the DEM-CFD coupling model established in this study could effectively and accurately capture the dynamic motions of soil and straw during a stubble-crushing operation. The simulation results showed that airflow substantially affected the soil and straw displacement and velocity. The airflow effect could accelerate the lateral and longitudinal throwing of the soil and the straw and expand the throwing range with little effect on their vertical position.

The longitudinal direction was the main throwing direction of soil and straw in the stubble-crushing operation. As the rotary speed increased, the lateral and longitudinal displacement and velocity of soil and straw increased evidently, which facilitated straw burial and soil layer exchange. The stubble-crushing blade rotary speed had no significant directional effect on the vertical motions of soil. After the stubble-crushing operation, the displacement of the shallow soil would fall and the displacement of the deep soil would rise, forming an exchange between the deep and shallow soil layers while achieving a certain degree of surface straw cover. The effect of the rotary speed on the lateral displacement and lateral velocity of the soil particles decreased with the increasing soil depth. After a stubble-crushing operation, the displacement of the shallow soil would fall and the displacement of the deep soil would rise, forming an exchange between the deep and shallow soil layers and realizing the change of soil layers. At the same time, most of the longitudinal soil was thrown further, which was shallow soil. In the field operation, the rotary speed should be increased appropriately to increase the lateral displacement of the straw, which in turn achieves the disturbance of the straw in the lateral direction. Taking into account the separation of the soil and straw and the formation of the desired seeding row, plain straight stubble-crushing blades with an angular speed of 540 rpm were able to form the optimal seeding row with a width of 80.86 mm. This also helps to achieve the separation of soil and

straw and creates good seed bed conditions. These results have implications for the future design of improved components and their evaluation for strip tillage.

**Author Contributions:** Conceptualization, methodology, software, Y.Y. and J.W.; validation, formal analysis, investigation, resources, data curation, Y.Y. and X.Z.; writing—original draft preparation, Y.Y. and J.W.; writing—review and editing, Y.Y. and S.Z.; supervision, S.Z. All authors have read and agreed to the published version of the manuscript.

**Funding:** This research was funded by the National Key Research and Development Program of China (Grant No. 2020YFD1000903).

**Institutional Review Board Statement:** Not applicable.

**Data Availability Statement:** Not applicable.

**Conflicts of Interest:** The authors declare no conflict of interest.

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
