# Peer review of "Effect of Rotary Speed on Soil and Straw Throwing Process by Stubble-Crushing Blade for Strip Tillage Using DEM-CFD"

_agriculture, doi:10.3390/agriculture13040877_

Round 1

Reviewer 1 Report

The article is well-written and could be accepted after minor corrections (in the text)

-Information about soil texture and moisture is also needed

-Information about means comparison is also needed

-Information about the travel speed of the tractor or planter is needed

-Information about the power requirement for the tractor is needed

Author Response

Dear Reviewer,

Thank you for your kindly comments and approval of the manuscript. According with your advice, we tried our best to amend the relevant part and made some changes in the manuscript. These changes will not influence the content and framework of the paper. All of your questions were answered below. And here we list the changes and marked in red in revised paper.

Comment 1: Information about soil texture and moisture is also needed.

Response: Thank you for your suggestion. The description of soil texture is on Line#250: “Soil in the Northeast is typically black soil with a clay loam texture containing 36.0% sand, 24.5% silt, and 39.5% clay [47].” For the convenience of readers, add relevant descriptions in the simulation experiment section based on your feedback. Please refer to the manuscript for details (Line#183“The black soil (clay loam) in Northeast China was used for this modeling.” Line#258“The average soil moisture content of the cultivation layer within 80mm of the experimental site was (20 ± 1) %, and average moisture content of maize root stubble is 184.3% (d.b).” and Table 1).

Comment 2: Information about means comparison is also needed.

Response: We thank for your comment. Based on your feedback, we have added relevant information on Line#294-296 of the manuscript: “the average velocity and displacement of four random soil particles at each depth and four random straw at the surface in different directions were taken for treatment.”

Comment 3: Information about the travel speed of the tractor or planter is needed.

Response: Information about the travel speed was written in the manuscript on Line#269: The forward travel speed was kept constant at 0.56 m×s-1, as was the rotor rotational speed at 810 rpm (equivalent to the rotational speed of the high-grade output shaft of the tractor). We apologize for any confusion you may have.

Comment 4: Information about the power requirement for the tractor is needed.

Response: We thank you for pointing this out. The tractor used in the field experiment was 45 horsepower. The relevant information has been added to the manuscript on Line#268.

Once again, thank you very much for your comments, suggestions, and taking the time to review our manuscript.

Yours Sincerely,

Yiwen Yuan

Reviewer 2 Report

1.     Please remove unnecessary hyphens, such as dis-placement, in-creasing, etc.

2.     Avoid bulk citations in the introduction [1-8]. Please only cite the relevant references.

3.     Please include the full definition of JKR (line#147), RNG (line#154).

4.     What are the assumptions made in the coupled DEM-CFD simulation? Please include it in the methodology.

5.     Briefly justify how and why the fixed time step was considered in the manuscript.

6.     How do authors determine the fixed time step size in the simulation?

7.     Include a figure to illustrate the principle coupling of DEM-CFD. For example, 1-way or 2-way.

8.     What kind of data (e.g., force, pressure, displacement, velocity) will be received by DEM and CFD in the coupling simulation?

9.     Please state clearly the definition of lower correlation (line#306). For example, provide the range R-square < 0.5 or else.

Author Response

Dear Reviewer,

Thank you for your kindly comments and suggestion concerning our manuscript. The comments and suggestions are all valuable and very helpful for revising and improving our paper, as well as the important guiding significance to our researches. We have studied comments carefully and have made correction which we hope meet with approval. All of your questions were answered below, and marked in red in revised paper.

Comment 1: Please remove unnecessary hyphens, such as dis-placement, in-creasing, etc.

Response: We apologize for the editorial error in the manuscript due to our carelessness. The corresponding errors in the manuscript have been corrected.

Comment 2: Avoid bulk citations in the introduction [1-8]. Please only cite the relevant references.

Response: We have streamlined the references based on your feedback, as detailed in the manuscript.

Comment 3: Please include the full definition of JKR (line#147), RNG (line#154).

Response: Thank you for your suggestion. The full definitions of the models used have been supplemented on Line#147“Hertz-Mindlin with JKR (Johnson-Kendall-Roberts) Cohesion contact model” and Line#165“RNG (Renormalization Group) k-ε turbulent flow energy dissipation rate model”.

Comment 4: What are the assumptions made in the coupled DEM-CFD simulation? Please include it in the methodology.

Response: Thanks for your comment, we have revised in 2.1. Development of coupling contact model part of the manuscript as: “k-ε turbulence model was developed based on Boussinesq’s assumption of isotropic eddy viscosity coefficient—Reynolds stress is proportional to the average velocity gradient.” on Line#163-165. Due to the mature application of the DEM-CFD coupling simulation method in agricultural machinery and other fields, and its extensive application in existing research, only a brief supplement is provided in Section 2.1.

Comment 5: Briefly justify how and why the fixed time step was considered in the manuscript.

Response: We thank you for raising this question. The time step is the time difference between every two calculations in the simulation calculation module. A fixed time step can maintain the continuity of the simulation process. Scholars have set fixed time steps in discrete element simulations, and to increase credibility, references have been added to the manuscript on Line#212—“Less than 20% Raileigh time step [43]”.

Comment 6: How do authors determine the fixed time step size in the simulation.

Response: We thank you for raising this question. As the time step directly affects the simulation time, it also affects the accuracy of the simulation. Based on the previous research results of the research group and in combination with other literature, the time step in this study was set to 8.2 × 10-5 s.

Comment 7: Include a figure to illustrate the principle coupling of DEM-CFD. For example, 1-way or 2-way.

Response: Thank you very much for your suggestion. “Figure 3. Flow chart of EDEM-FLUENT coupling processes and information exchange between two solvers.” has been added to the manuscript to illustrate the principle coupling of DEM-CFD.

Comment 8: What kind of data (e.g., force, pressure, displacement, velocity) will be received by DEM and CFD in the coupling simulation?

Response: We thank you for raising this question. We completed the simulation experiment by coupling the Coupling Server module in EDEM software with Fluent software through the udf interface. The displacement and velocity data obtained from the experiment are directly exported and processed through the EDEM post-processing module.

Comment 9: Please state clearly the definition of lower correlation (line#306). For example, provide the range R-square < 0.5 or else.

Response: Thank you very much for your suggestion. The meaning conveyed by the textual expressions in the original manuscript is ambiguous. To increase the readability of the article, we have made modifications to the corresponding content in the manuscript, as detailed on Line#329. “The overall lower determination coefficient (R2) value of the longitudinal motion of straw than that of soil is due to the instability of straw throwing during stubble-crushing blade operation.”

Once again, thank you very much for your comments, suggestions, and taking the time to review our manuscript.

Yours Sincerely,

Yiwen Yuan
